# The bandgap-detuned excitation regime in photonic-crystal resonators

Yan Jin [1,2] ✉, Erwan Lucas [3], Jizhao Zang [1,2], Travis Briles[1], Ivan Dickson[1,2], David Carlson[4] & Scott B. Papp [1,2]

Control of nonlinear interactions in microresonators enhances access to classical and quantum field states across nearly limitless bandwidth. A recent innovation has been to leverage coherent scattering of the intraresonator pump field as a control of group-velocity dispersion and nonlinear frequency shifts, which are precursors for the dynamical evolution of new field states. Yet, since nonlinear-resonator phenomena are intrinsically multimode and exhibit complex modelocking, here we demonstrate a new approach to controlling nonlinear interactions with bandgap modes completely separate from the pump laser. We explore this bandgap-detuned excitation regime through generation of benchmark optical parametric oscillators (OPOs) and soliton microcombs. Indeed, we show that mode-locked states are phase matched more effectively in the bandgap-detuned regime in which we directly control the modal Kerr shift with the bandgaps without perturbing the pump field. In particular, bandgap-detuned excitation enables an arbitrary, mode-by-mode control of the backscattering rate as a versatile tool for mode-locked state engineering. Our experiments leverage nanophotonic resonators for phase matching of OPOs and solitons, leading to control over threshold power, conversion efficiency, and emission direction that enable application advances in high-capacity signaling and computing, signal generation, and quantum sensing.

Lasers are versatile and precise sources with diverse applications across a range of industries and scientific disciplines, contributing to advanced optical communication[1], atomic and molecular spectroscopy[2], and quantum technologies[3]. The capability to generate laser sources with a nearly arbitrary spectrum and quantum-limited noise properties is an enduring challenge that stimulates exploration of new light-matter interactions. Integrated nonlinear photonics plays a key role in enabling the spectrum of light to be dramatically transformed from the input to the output of a device. In particular, nonlinear waveguides transform a mode-locked laser into supercontinuum outputs with exceptionally broad bandwidth at low operating power[4–6] and with complex mixtures of nonlinear interactions[7]. Kerr microresonators enable the generation of vastly broadband optical parametric oscillators (OPOs) and soliton microcombs through the conversion of a continuous-wave (CW) pump laser, according to a complex mixture of resonance conditions, group-velocity dispersion of the resonator modes, and the intraresonator pump field[8–16].

Control of nonlinear interactions in microresonators offers the opportunity to engineer new laser sources[17]. Group-velocity dispersion (GVD, hereafter dispersion) engineering has been an important advance, enabling direct spectrum control in microcombs[18,19]. Adjusting the pump laser power and detuning also directly controls a soliton microcomb[20]. Given that solitons are isolated excitations of the microresonator, interacting soliton ensembles offer unique properties[21].

[1]Time and Frequency Division, National Institute of Standards and Technology, Boulder, CO, USA. [2]Department of Physics, University of Colorado, Boulder, CO, USA. [3]Laboratoire ICB, UMR 6303 CNRS-Université de Bourgogne, Dijon, France. [4]Octave Photonics, Louisville, CO, USA. ✉e-mail: yan.jin@colorado.edu

Moreover, ensembles of coupled microresonators enable fundamentally new interactions, including solitons that coexist in several devices[22]. Direct modification of the intensity-dependent refractive index is also a natural tool for controlling solitons[23,24]. Among the methods that have been explored to manipulate microcombs, a recent innovation has been photonic-crystal ring resonators (PhCRs). These employ a sub-wavelength nanostructure to couple forward and backward propagation through backscattering, inducing mode-frequency splittings associated with the backscattering rate[18,25].

PhCRs offer direct control of the phase matching and nonlinear dynamics of microresonators. In particular, pump-laser excitation of the split mode in PhCRs allows universal phase matching for four-wave mixing in both the normal and anomalous dispersion regimes, enabling OPOs[26,27], bright solitons in anomalous dispersion[25], access to the dark-to-bright soliton continuum in normal dispersion[28], and use in nonlinear resonator circuits to optimize microcomb performance metrics[9]. Recently, PhCR bandgaps for unpumped modes have been used to define the output wavelength in OPOs[29] and induce interesting microcomb dynamics[18,30]. Although PhCRs intrinsically program mode-by-mode dispersion, since the nanostructured waveguide defines the optical mode, other approaches, including coupled resonators, enable similar controls[31,32]. Still, the use of coherent backscattering in PhCRs opens excess optical loss channels, particularly since practical devices operate in the large bandgap limit where forward and backward propagation are strongly coupled. These devices include specific limitations, such as increased threshold power, reduced conversion efficiency, and predominant emission of newly generated OPO and soliton light in the backward direction, i.e., emitted toward the pump laser.

Here, we explore the bandgap-detuned excitation regime of PhCRs in which we open optical bandgaps, mode-detuned from the pump laser. Operating PhCRs in this regime avoids splitting the pump amongst the forward and backward directions, yet we preserve the capability for universal phase matching. Moreover, by designing bandgaps to interact directly with mode-locked nonlinear states of the resonator, we exploit the inherent multimode composition of target states in their construction. The ratio of forward and backward coupling, and hence the effective excess loss toward the backward direction, is tunable in the bandgap-detuned regime. Through this effect, we introduce a new framework for nonlinear-state engineering based on an effective integrated dispersion parameter, which directly characterizes phase matching of field states. We explore benchmark OPOs and soliton microcombs in bandgap-detuned PhCRs, demonstrating that these states are more effectively phase-matched and can be created with reduced threshold power, higher efficiency, and the control to realize predominantly forward emission. Furthermore, we demonstrate a design process for OPO lasers and soliton microcombs that could be used as sources for applications.

## Results

A PhCR is a three-port nonlinear system with an input port receiving the pumping field $F$, and two output ports with modal fields $A_\mu$ and $B_\mu$ propagating respectively in the forward direction (i.e., the same as the pump) and the backward direction, where $\mu$ is the mode number with respect to the pumped mode. At a designated mode $\mu_s$, coherent backscattering from the periodic, sub-wavelength pattern of the PhCR lifts the initial degeneracy of $A_{\mu_s}$ and $B_{\mu_s}$, leading to a red-shifted (lower frequency) and a blue-shifted (higher frequency) resonance, separated by a bandgap $\Gamma_{\mu_s}$. The dispersion of the resonance frequencies is perturbed and follows the integrated dispersion relation $D_{\text{int}}(\mu) \equiv \omega_\mu - \omega_0 - \mu D_1 = D_2\mu^2/2 \pm \Gamma_\mu/2$, where $\omega_\mu$ is the resonance frequencies of the ring resonator, $\omega_0$ is the pump mode frequency, $D_1/2\pi$ is the free spectral range, $D_2$ is the second-order dispersion coefficient, and $\Gamma_\mu = 0$ if $\mu \neq \mu_s$. To describe the dynamics of the PhCR, we introduce

the coupled-mode LLEs written in the spectral domain[33]:

$$\dot{A}_\mu = -(1+i\alpha)A_\mu - iD_2\mu^2 A_\mu/\kappa + i\Gamma_\mu B_\mu/\kappa + i\mathcal{F}(|A|^2 A)_\mu + i2A_\mu P_b + F\delta_{\mu,0} \tag{1}$$

$$\dot{B}_\mu = -(1+i\alpha)B_\mu - iD_2\mu^2 B_\mu/\kappa + i\Gamma_\mu^* A_\mu/\kappa + i\mathcal{F}(|B|^2 B)_\mu + i2B_\mu P_a, \tag{2}$$

where $F$ refers to the normalized pump field, $\alpha$ is the detuning, $\delta_{\mu,0}$ is the Kronecker symbol, $\kappa$ is the linewidth, $\Gamma_\mu/\kappa$ is the normalized half-bandgap in mode $\mu$, $P_a = \sum_\eta |A_\eta|^2$, $P_b = \sum_\eta |B_\eta|^2$, and $\dot{A}_\mu = dA_\mu/d\tau$ where $\tau = t\kappa/2$ is the normalized time, and $\mathcal{F}(\cdot)_\mu$ represents the Fourier operator. Our framework for nonlinear-state engineering in PhCRs is based on an effective integrated dispersion $\tilde{D}_{\text{int}}(\mu)$, which characterizes the frequency mismatch of the field states,

$$\tilde{D}_{\text{int}}(\mu) \equiv \text{Re}\left(i\frac{dA_\mu/dt}{A_\mu}\right) = \frac{\kappa}{2}\text{Re}\left(i\frac{\dot{A}_\mu}{A_\mu}\right) = \alpha\frac{\kappa}{2} + \frac{D_2}{2}\mu^2$$
$$+ \epsilon_\mu - \Delta_\mu - P_b\kappa - \delta_{\mu,0}\,\text{Im}\left(\frac{F}{A_0}\right)\frac{\kappa}{2} \tag{3}$$

where $\Delta_\mu \equiv \text{Re}\,(\mathcal{F}(|A|^2 A)_\mu/A_\mu)\kappa/2$ is the modal Kerr shift[25], and $\epsilon_\mu \equiv \text{Re}\,(-\Gamma_\mu \frac{B_\mu}{A_\mu})/2$ is the bandgap-induced frequency shift. The three main contributors in Equation (3) are the background dispersion $D_2\mu^2/2$, the Kerr nonlinearity $\Delta_\mu$, and the mode shift $\epsilon_\mu$ due to coupling between the counterpropagating fields. For $F = 0$, $\Delta_\mu = 0$, $\epsilon_\mu = \pm\Gamma_\mu/2$ indicates two components of the split mode, one blue-shifted mode ($\epsilon_\mu > 0$) and the other red-shifted mode ($\epsilon_\mu < 0$), and $\tilde{D}_{\text{int}}(\mu)$ reduces to the dispersion of the resonator modes $D_2\mu^2/2 \pm \Gamma_\mu/2$. The different signs of $\epsilon$ indicate the frequency of the comb line at $\mu$ is greater ($\epsilon_\mu > 0$) or smaller ($\epsilon_\mu < 0$) than the center of the two resonances at the split mode. In mode-locked states, $\tilde{D}_{\text{int}}(\mu)$ is linear with $\mu$, denoting a stable and stationary field configuration. Therefore, we express $\tilde{D}_{\text{int}}(\mu) = \mu\delta\omega_{\text{rep}}$, where $\delta\omega_{\text{rep}}/2\pi = f_{\text{rep}} - \text{FSR}$ is the difference between the repetition rate of the comb and the free spectral range, which is measured in the cold resonator. In mode-locked states, where $\tilde{D}_{\text{int}}(\mu)$ and $D_2\mu^2/2$ remain constant, the relative field amplitudes $A_\mu$ and $B_\mu$ are influenced by the interplay between the bidirectional coupling strength $\epsilon_\mu$ and the Kerr nonlinearity $\Delta_\mu$.

In the bandgap-detuned regime, we implement nonlinear state engineering by introducing bandgaps in unpumped modes. First, we consider the case $\Gamma_\mu = \Gamma\delta_{\mu,\mu_s}$, where $\mu_s \neq 0$ is the split mode number and $\delta$ is the Kronecker delta. Due to the constraint of $\tilde{D}_{\text{int}}$ for a mode-locked state, we can control each of the field amplitudes $A_\mu$ or $B_\mu$ with each bandgap $\Gamma_\mu$, which sets the bandgap-induced frequency shift $\epsilon_\mu$. This arbitrary, mode-by-mode control parameter also sets the excess power "lost" to backscattering in the PhCR to directly shape nonlinear states. Furthermore, in the bandgap-detuned regime, mode-locked states are excited in either the red mode ($\epsilon_\mu < 0$) or the blue mode ($\epsilon_\mu > 0$), allowing another new facet to control such states. In contrast, in the conventional regime ($\Gamma_\mu = \Gamma\delta_{\mu,0}$) where the red-shifted resonance of the split mode is pumped to generate OPOs or solitons[25], only $\epsilon_0 < 0$ is allowed.

To understand nonlinear-state excitation in this new scenario, we present benchmark simulations of OPO states in the conventional (Fig. 1a) and bandgap-detuned (Fig. 1b) regimes to directly show the advantages of the latter regime. These simulations compare the spectra of forward ($|A_\mu|^2$) and backward ($|B_\mu|^2$) emission and the intraresonator pulse profiles. We set $F$ as 2.6 and 2, and $\mu_s$ as 0 and 10 in the simulation for Fig. 1a, b, respectively. The forward spectra or pulses are in blue, and the backward are in cyan. In conventional PhCRs, the generated OPO light propagates backward, while in the bandgap-

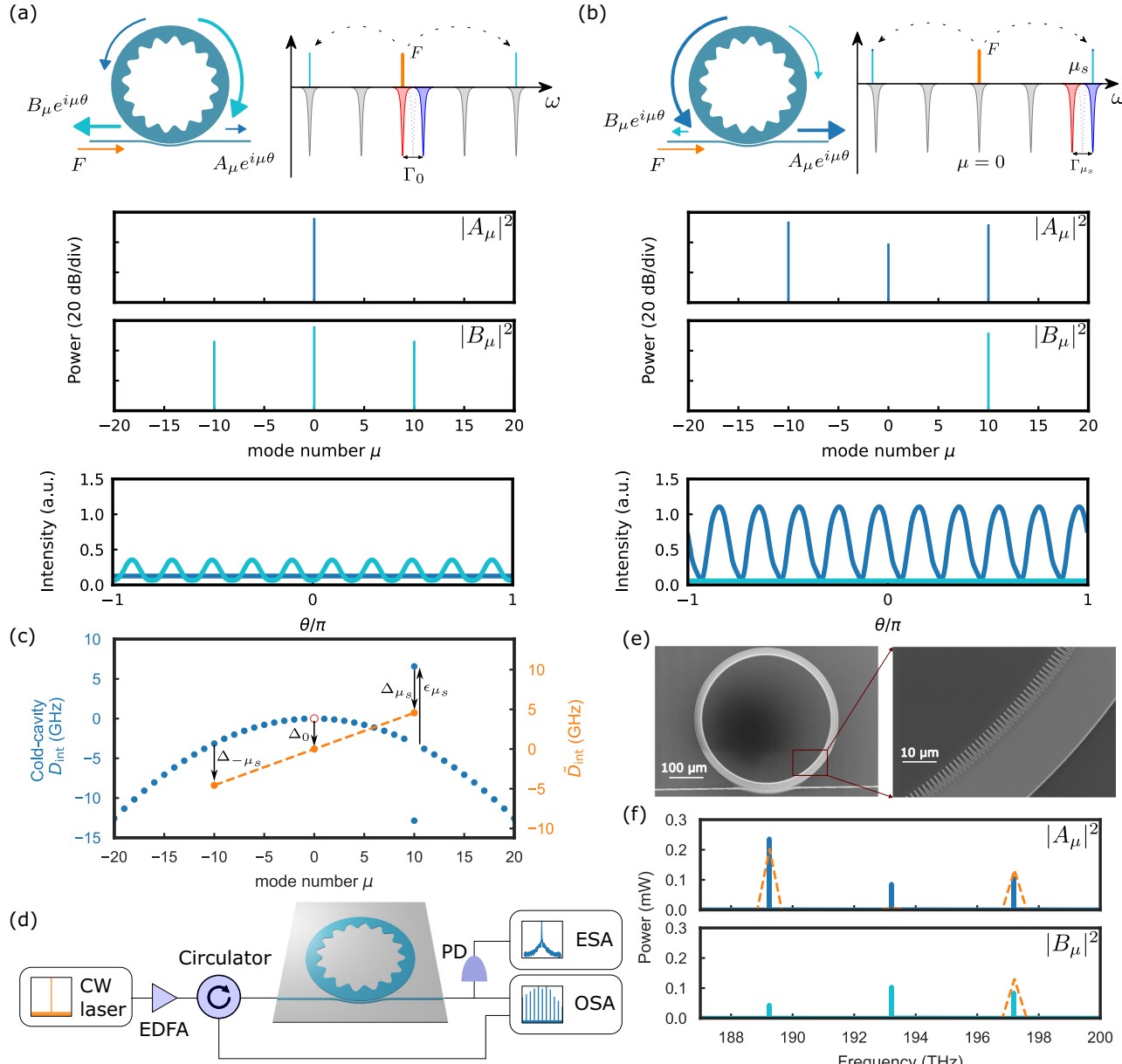

**Fig. 1 | Benchmark OPOs. a** Conventional PhCR OPO ($\mu_s = 0$), where the red-shifted mode is pumped and the combs propagate in the backward direction against the pump. **b** Bandgap-detuned PhCR OPO ($\mu_s = 10$), where the combs mainly propagate in the forward direction, i.e, the same direction as the pump. Both (**a**) and (**b**) present the ring structure, $D_{int}$ normalized to $\kappa/2$, mode structure, the forward (blue) and backward (cyan) spectra with comb lines at $\mu = 10$, and pulse intensity profiles normalized to $F^2$. **c** Effective integrated dispersion $\tilde{D}_{int}$ for $\mu_s = 10$ in the cold cavity (blue) and hot cavity (orange). As we increase the pump power, the modal Kerr shift $\Delta_\mu$ compensates the original cold cavity dispersion $D_{int}$ (blue) and the bandgap-induced shift $\epsilon_{\mu_s}$, and pushes the optical modes to a straight line (orange), which achieves the phase-matching condition and generates the stable OPOs. **d** Setup. EDFA: Erbium-doped fiber amplifier. OSA: optical spectrum analyzer. ESA: electrical spectrum analyzer. PD: photodiode. **e** SEM images for PhCR. The ring resonator designs shown in this paper are exaggerated so that the SEM images are more clear. **f** Forward and backward spectra for $\mu_s = 10$ for simulation (orange dashed lines) and experiment (solid lines). Pump reflection from the input facet is anomalously large, since waveguide coupling loss does not occur. We calibrate this effect by tuning the pump off resonance.

detuned regime it propagates predominantly forward. This difference emerges because at the parametric threshold, the backward gain exceeds the forward gain in the conventional regime, while the bandgap-detuned regime is characterized by the dominance of the forward gain. These results, based on the solution of the highly accurate LLE, demonstrate the enhanced properties of the bandgap-detuned regime; see Supplementary Information for details of the parametric gain. Due to a stronger backward pump field in the conventional regime, the OPOs only propagate in the backward direction. In the bandgap-detuned regime, because of the coupling between the forward and backward fields at $\mu_s = 10$ induced by the bandgap, there is

a backward-propagating OPO line only at $\mu = 10$. The pulse intensities in Fig. 1a, b present the corresponding $|A(\theta)|^2$ (blue) and $|B(\theta)|^2$ (cyan) normalized to the pump power $F^2$ in arbitrary units (a.u.). They all have 10 oscillations in a round trip because the OPO lines emerge at $\mu = \pm 10$ for both regimes. Comparing these two normalized intensity profiles, it is clear that the bandgap-detuned regime has a higher efficiency.

Since the constraint on $\tilde{D}_{int}(\mu)$ is the basis for nonlinear-state engineering, we show in Fig. 1c that this quantity explains the formation and advantages of OPOs in the bandgap-detuned regime. For $F = 0$, we plot the cold-cavity effective dispersion (blue) $\tilde{D}_{int}(\mu) = D_{int}(\mu) = D_2\mu^2/2 \pm \Gamma_\mu/2$, and the pump mode is indicated by

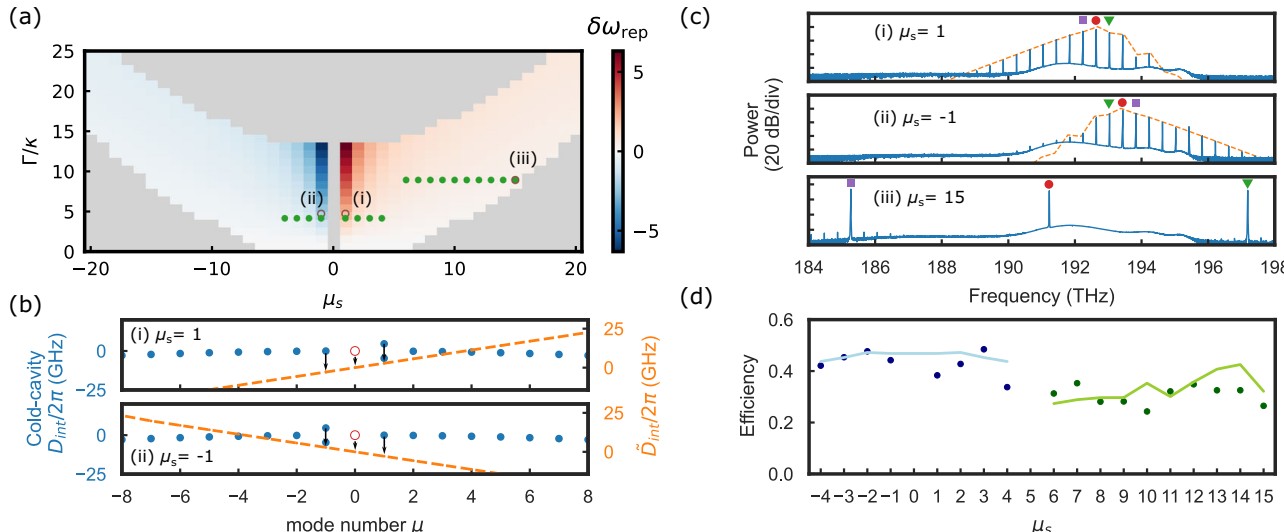

**Fig. 2 | Existence map and the corresponding $D_{int}$, spectra, and efficiency for the measured devices. a** Existence map with $F = 2$, $D_2/\kappa = -0.0185$. The colors indicate the values of $\delta\omega_{rep}$ as the signal and idler are excited, the brown empty circles marked with (i–iii) are explained in (**b**) and (**c**), and the green dots refer to the efficiency data points in (**d**). **b** Phase matching diagrams with (i) $\mu_s = 1$ and (ii)

$\mu_s = -1$. **c** Simulated (orange) and measured (blue) spectra with (i) $\mu_s = 1$, (ii) $\mu_s = -1$, and (iii) $\mu_s = 15$. Red circle: pump, green triangle: split mode at $\mu_s$, purple square: mode at $-\mu_s$. **d** Forward efficiency of two devices (blue and green) as we pump different modes. The solid lines are the simulated efficiency, and the dots are the efficiency measured in experiments.

the red circle. The split mode at $\mu_s = 10$ with 2 resonances indicates that there is a coupling between the forward and backward fields, which results in generating both the forward and backward OPO or comb lines at that mode. In the experiment, we measure $D_{int}(\mu)$ in the forward direction; see "Methods" section for more details. Since self-phase modulation is half of cross-phase modulation in the flat state, the modal Kerr shift at $0, \pm\mu_s$ obeys $\Delta_0 < \Delta_{\mu_s}, \Delta_{-\mu_s}$. Thus, the blue mode at $\mu_s$ is excited so that $\epsilon_{\mu_s} > 0$, and $\tilde{D}_{int}(\mu)$ at $\mu = -\mu_s, 0, \mu_s$ connect directly in a straight-line fashion (orange dashed line). This characteristic stands in contrast to the conventional regime, where the red mode of the pumped split mode is excited ($\epsilon_0 < 0$). The split mode $\mu_s$ targets the frequency of the signal or idler, and $\epsilon_{\mu_s} > 0$ increases $\Delta_{\mu_s}$. The increased Kerr shift $\Delta_{\mu_s}$ implies that higher power accumulates in the target mode $\mu_s$ and the symmetric mode $-\mu_s$, which explains enhanced conversion efficiency ("Methods" section) in the bandgap-detuned OPOs.

We explore the generation of OPOs and soliton microcombs throughout this paper with the experimental setup in Fig. 1d. We generate the pump with a continuous-wave (CW) laser, amplified by an erbium-doped fiber amplifier (EDFA), and we couple this light into a tantala ($Ta_2O_5$) PhCR[34,35]. An optical circulator separates the forward propagating pump and the backward propagating field from the resonator. We measure both the forward- and backward-propagating fields with an optical spectrum analyzer (OSA), and we use an electrical spectrum analyzer (ESA) to measure the relative intensity noise of the forward spectra. We show the physical structure of the PhCR with scanning electron microscope (SEM) images in Fig. 1e. The outer sidewall of the ring is circular, while we create a spatial modulation on the inner wall of the ring by a periodic nanostructure; see Methods.

To generate bandgap-detuned OPOs, we tune the pump laser frequency to a mode of the PhCR and monitor the device output. Our framework provides a specific expectation for the resulting state. Figure 1f shows the simulated and experimental spectra for case $\mu_s = 10$, highlighting the close correspondence of the traces. The presented experimental spectra are measured with the OSA, and the on-chip power is 14 dB greater, including 4 dB for the chip-to-fiber coupling and a 10 dB attenuation. In the forward direction, both the simulation and the experimental data corroborate that the pump is almost fully depleted, with its power reduced below that of the signal

and the idler. In our simulation, only the comb mode at $\mu = \mu_s$ is non-zero in the backward direction, but in the experiment, the backward comb lines at $\mu = 0, -\mu_s$ also appear due to the reflection of the angled waveguide facets at the edges of the chip. The backward comb line at $\mu = 0$ is particularly strong because the pump is reflected at both the front (input) and rear (output) facets of the chip, while the comb line at $-\mu_s$ is only reflected at the rear facet of the chip. The reflection at the front facet is much stronger for the pump because, at this point, the pump hasn't been attenuated yet. In our simulation, these comb lines don't show up because Equations (1–2) only describe the dynamics of the fields inside the ring cavity.

The central characteristic that controls the generation of bandgap-detuned OPOs is the mode number of the bandgap relative to the pump mode $\mu_s$. In particular, we explore phase matching with specific parameters ($F$, $d_2$, $\mu_s$, and normalized half-bandgap $\Gamma_\mu/\kappa$) that enable OPO generation. Figure 2a presents the OPO existence map with respect to the split-mode number $\mu_s$ and the bandgap strength $\Gamma_{\mu_s}/\kappa$ for given driving $F = 2$ and dispersion $D_2/\kappa = -0.0185$. The gray region indicates the area in which no OPO is generated. The colored region gives the values of $\delta\omega_{rep}$ that characterize the difference between $f_{rep}$ and FSR. Note that this map was not computed via LLE simulations, but by exploring the steady state of the pump mode ($a_0$) and evaluating the dependence of the parametric gain at the modes $A_{\mu_s}$, $A_{-\mu_s}$ and $B_{\mu_s}$ as a function of $\mu_s$ and $\Gamma/\kappa$; see Supplementary Information for detailed derivation. The brown circles (marked with i–iii) and green dots are experimentally measured data points for devices with different $\mu_s$ and $\Gamma/\kappa$, which represent our observation points in Fig. 2b–d.

In the bandgap-detuned mode-locked regime, the unpumped split modes make the integrated dispersion asymmetric, leading to a non-zero slope coefficient $\delta\omega_{rep}$ in $\tilde{D}_{int}(\mu)$. The mode detunings and the strength of the bandgaps determine the sign of $\delta\omega_{rep}$. For OPOs, if $\mu_s > 0$, then $\delta\omega_{rep} > 0$ and vice versa. This is illustrated by the phase matching diagrams in Fig. 2b; for simplicity, we take (i) $\mu_s = 1$ and (ii) $\mu_s = -1$. For either case, the blue mode at $\mu = \mu_s$ is excited. When the Kerr shift compensates for the original dispersion, if $\mu_s = 1$, then $\epsilon_1 > 0$, and the slope of $\tilde{D}_{int}(\mu)$ is greater than 0; while if $\mu_s = -1$, then $\epsilon_{-1} > 0$ and the slope is negative. To verify this, we select a device with FSR = 398.078 GHz and pump it into the modes adjacent to the split

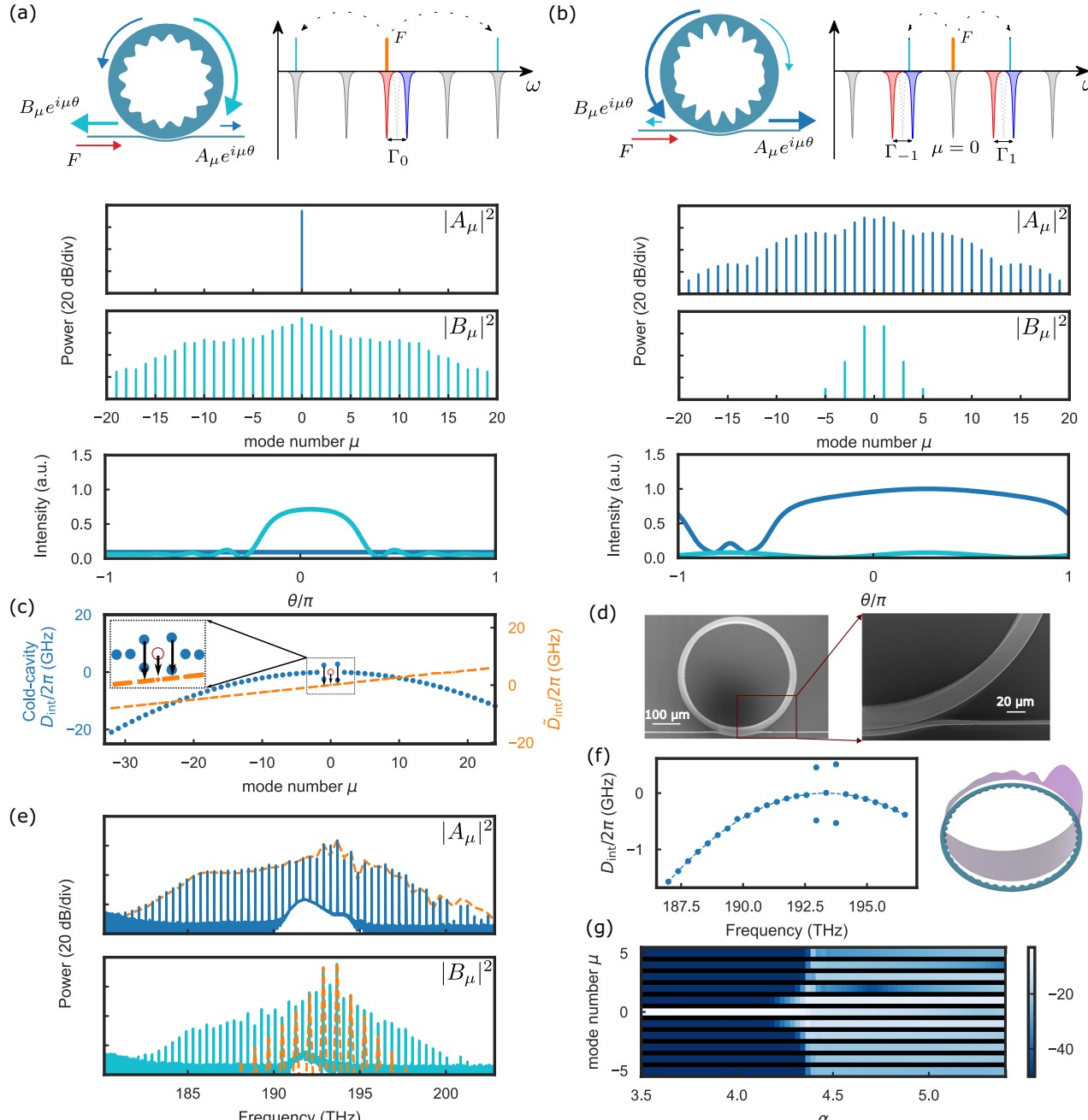

**Fig. 3 | Benchmark soliton microcombs. a** Conventional regime of generating combs in a photonic-crystal ring resonator. The red mode of the pump mode is on resonance, and the backward combs dominate. **b** Bandgap-detuned excitation regime. Modes $\mu_s = \pm 1$ are split, and the simulated spectra and pulse indicate the forward-propagating combs. In (**a**, **b**), the blue traces indicate the forward spectra or pulse, and the cyan traces indicate the backward spectra or pulse. **c** Phase-matching diagram for the bandgap-detuned regime in the cold and hot cavities. **d** SEM images. **e** Experimental (solid) and simulated (orange dashed) spectra in the forward ($|A_\mu|^2$) and backward ($|B_\mu|^2$) direction for the bandgap-detuned regime. **f** Measured dispersion and simulated pulse on the ring resonator. **g** Simulated spectra with respect to $\alpha$. The unit of the map is dB.

mode so that $\mu_s$ can be 1 or −1. The cold cavity dispersion $D_{int}$ displayed in Fig. 2b corresponds to the cases $\mu_s = \pm 1$ for this device, and the first two panels in Fig. 2c show the corresponding spectra when we pump these different split modes. The green triangles point to the split mode $\mu_s$, the red circles indicate the pump mode, and the purple squares indicate the mode at $-\mu_s$. We then measure the repetition rates (line spacing) and find $f_{rep} = 398.488$ GHz for $\mu_s = 1$, and $f_{rep} = 397.569$ GHz for $\mu_s = −1$. This corresponds to $\delta\omega_{rep}/2\pi = 410$ MHz ($\mu_s = 1$), and $\delta\omega_{rep}/2\pi = −509$ MHz ($\mu_s = −1$). The difference in the values of $\delta\omega_{rep}$ is due to different linewidths of the pump mode for $\mu_s = 1$ or −1. In our simulation, we found $\delta\omega_{rep}/2\pi = 392$ MHz and −459 MHz, respectively, which

is close to our experimental results. It validates our prediction of the behavior of $\tilde{D}_{int}$ in Fig. 3 and showcases the precise control of nonlinear interaction in the bandgap-detuned regime.

In addition to nonlinear-state engineering, which involves controlling the relative amplitude of $A_\mu$ and $B_\mu$, the bandgap-detuned regime provides the capability to efficiently generate forward-propagating OPOs with varying bandwidth, while minimizing excess loss resulting from backward reflection. We select a device with a split mode wavelength around 1550 nm and $\mu_s \in \{ \pm 1, \pm 2, \pm 3, \pm 4 \}$, and another device with a split mode around 1521 nm and $\mu_s \in [6, 15]$. Their locations on the existence map are marked with green dots in Fig. 2a. In

particular, the data point with $\mu_s = 5$ lies at the boundary of the existence map and is marked with a brown circle, showing the prediction ability of the existence map. We plot its forward spectra in panel (iii) of Fig. 2c. The pump mode is well depleted and has a lower power than the signal and idler, indicating good efficiency performance. Figure 2d shows the efficiency of the two devices in the forward direction as a function of $\mu_s$, obtained by pumping different modes. See "Methods" section for more details about the OPO conversion efficiency.

We now turn to the generation of soliton microcombs in normal dispersion with the bandgap-detuned scheme. By designing PhCRs with multiple unpumped bandgaps, we open up mode-locked-state engineering of solitons, following the same analysis based on the frequency mismatch operator $\tilde{D}_{\mathrm{int}}$ of the fields in the device. We begin by comparing soliton generation in PhCRs with conventional (Fig. 3a) and bandgap-detuned (Fig. 3b) excitation. With the same pump power setting $F^2$, we calculate the evolution of the forward (blue) and backward (cyan) propagating comb as the pump tunes into the mode $\mu = 0$, into the lower frequency resonance in the conventional case ($\Gamma_0 > 0$), or to a bandgap-detuned scenario with equal symmetric bandgaps $\Gamma_{\pm 1}$ at $\mu_s = \pm 1$. We obtain the comb spectra $|A_\mu|^2$ and $|B_\mu|^2$, and the corresponding intraresonator temporal profiles in the forward (blue) and backward (cyan) directions. In the conventional PhCR scheme, the intracavity field evolves from primary OPO sidebands to a soliton microcomb in the backward direction. In the case of bandgap-detuned PhCR, parametric gain arises in the split modes $\mu_s$, resulting in the direct generation of sidebands. This initial modulation further evolves into a deterministic, forward-propagating dark soliton, made possible by the lack of splitting of the pumped mode, ensuring that most of its power propagates forward. Conversely, the backward-coupled comb modes at $\mu_s = \pm 1$ give rise to a weaker 2-FSR spaced comb in this direction.

To understand phase matching and the formation of bandgap-detuned soliton microcombs, we plot $\tilde{D}_{\mathrm{int}}(\mu)$ in Fig. 3c. Considering an example where the bandgaps at $\mu_s = \pm 1$ are not equal ($\Gamma_1 > \Gamma_{-1}$) highlights an important practical element of bandgap-detuned microcombs[30]. In this case, $D_{\mathrm{int}}(\mu)$ is plotted in blue, and the red circle indicates the pump mode. At threshold, the modal Kerr shift obeys $\Delta_1 \simeq \Delta_{-1} > \Delta_0$ because cross-phase modulation is larger than self-phase modulation, which is shown by the black arrow. As we increase the pump power ($F = 2$ in this case), the modal Kerr shift $\Delta_\mu$ compensates for the dispersion $D_{\mathrm{int}}$ with the help of the mode splittings at $\mu_s = \pm 1$, thus the phase matching is achieved and $\tilde{D}_{\mathrm{int}}(\mu)$ evolves into a straight line (orange). The mode in $\mu = 1$ is more blue-shifted than in $\mu = -1$, therefore, $\epsilon_1 > \epsilon_{-1}$, thus $\tilde{D}_{\mathrm{int}}(\mu)$ exhibits a positive slope in the mode-locked state, increasing the repetition rate of the comb ($\delta\omega_{\mathrm{rep}} > 0$).

To experimentally explore the generation of bandgap-detuned dark solitons, we study the PhCR device in Fig. 3d with bandgap modes $\mu_s = \pm 1$. We characterize $D_{\mathrm{int}}$ of the fabricated device by a calibrated frequency scan of our tunable laser across several resonator modes; see Fig. 3f. In particular, we resolve the resonator dispersion and the bandgaps $\Gamma_1/2\pi = 1.04$ GHz, $\Gamma_{-1}/2\pi = 0.94$ GHz, $D_1/2\pi = 397.848$ GHz, and $D_2/2\pi = -12.3$ MHz. On the basis of this information, we simulate the intraresonator intensity pattern of the bandgap-detuned soliton. Experimentally, we pump this PhCR with a power corresponding to $F = 2.5$ and slowly tune the pump laser frequency on resonance, starting from the blue-detuned side. This procedure yields a soliton microcomb propagating in the forward direction, which we characterize by measuring the optical spectrum of the forward (solid blue line) and backward (solid teal line) outputs from the device. Comparisons to our simulation (orange line in Fig. 1e) show a good match between the predicted forward comb and the experiment, confirming the close connection between the designed bandgaps, our mode-locked state engineering with $\tilde{D}_{\mathrm{int}}(\mu)$, and the operational generation of solitons in such devices. As we predict from $\tilde{D}_{\mathrm{int}}(\mu)$ in Fig. 3c, the larger bandgap at $\mu = 1$ than $\mu = -1$ leads to $\delta\omega_{\mathrm{rep}} > 0$, so there are more

modes with positive $\Delta_\mu$ for $\mu < 0$ than $\mu > 0$, leading to an asymmetric comb with a flatter comb profile for $\mu < 0$, as shown in the forward combs in Fig. 3e. Indeed, we measured $\delta\omega_{\mathrm{rep}}/2\pi$ to be 21 MHz, consistent with our prediction that $\delta\omega_{\mathrm{rep}} > 0$ in this device. Regarding the measured spectra and simulation in the backward direction, chip facet reflections obscure the observation of the entire backward spectrum. Nonetheless, characteristic spectral peaks in the modes $\pm 1$ are evident. Figure 3g shows the simulated spectra, focusing on the first few lines, upon scanning the detuning $\alpha$. It verifies our prediction that the comb lines at $\mu_s = \pm 1$ are generated first and then trigger the development of the entire comb, which is different from traditional soliton generation, where the primary combs are formed away from the pump.

Expanding on this framework for mode-locked-state engineering of dark solitons, we explore other states by way of design with $\tilde{D}_{\mathrm{int}}$. Important examples include soliton crystals[21] and bright pulses in normal dispersion[28]. To create a soliton crystal in the bandgap-detuned regime, we place bandgaps at $\mu_s = \pm 3$. Figure 4a shows the phase-matching diagram with the dispersion of the designed resonator and the simulated $\tilde{D}_{\mathrm{int}}$ when soliton crystals are generated. We show an exaggerated PhCR nanostructure with high-resolution SEM, and we characterize $D_{\mathrm{int}}$ with a calibrated laser frequency scan. To generate soliton crystal microcombs, we pump the device at the level of $F = 2.4$ by scanning the laser frequency into the mode $\mu = 0$. Figure 4b shows spectra and soliton-crystal formation at an initial detuning setting (i) and then at a different value of $\alpha$ (ii), where the laser frequency was slightly reduced. Because phase matching is designed to occur on modes $\mu_s = \pm 3$, the initial modulation pattern (i) is composed of three periods in the resonator, leading to the formation of a soliton crystal with three pulses of equal temporal space in the resonator[21]. When the laser frequency is decreased, the pulse spacing is perturbed, leading to the appearance of natively spaced lines in the spectra (see panel (ii)), as observed in soliton crystal combs with a defect[21]. We attribute this phenomenon to the interaction between the oscillating tails of the switching waves that form dark pulses[36].

We can form bright pulses in the bandgap-detuned regime by splitting many modes of the PhCR except for the pump mode, as shown in Fig. 4c. These split modes individually control the relative amplitude of $A_\mu$ and $B_\mu$, creating the dispersion profile needed for bright solitons and controlling the excess loss to coherent backscattering. We study an example in which twenty modes $|\mu_s| \leq 10$, except the pump, are split by an equal amount to create a so-called meta-dispersion in the mode spectrum[18]. Despite the large number of split modes, the principle for phase matching remains the same, and more phase matching pathways are enabled. At the parametric oscillation threshold, the entire higher frequency branch of the $D_{\mathrm{int}}$ spectrum is shifted to a lower frequency due to the Kerr effect. Therefore, a soliton forms as the pump mode and the blue-shifted branch align, creating a $\tilde{D}_{\mathrm{int}}$ profile that is a straight line. The associated photonic crystals form a more intricate corrugated pattern in the ring resonator, as shown in the SEM image in Fig. 4c. The measured dispersion verifies that the targeted mode structure is implemented correctly and that the pump mode is not split. Interestingly, this split-mode distribution consistently creates a single pulse in the ring resonator. Figure 4d presents the spectra from simulation and experiments, which allow us to characterize the generated state with respect to the $\tilde{D}_{\mathrm{int}}$ design. Despite an increased fraction of the comb power emitted in the backward direction because multiple modes are split, more power accumulates in the modes $|\mu| \leq 10$ and creates a bright pulse.

## Discussion

In summary, we have explored the bandgap-detuned excitation regime of PhCRs, demonstrating modelocked state engineering by design of modal Kerr shifts. The bandgap-detuned regime supports a rich state space of unique OPOs and soliton microcombs that offer low threshold power, high efficiency, and control over forward and backward

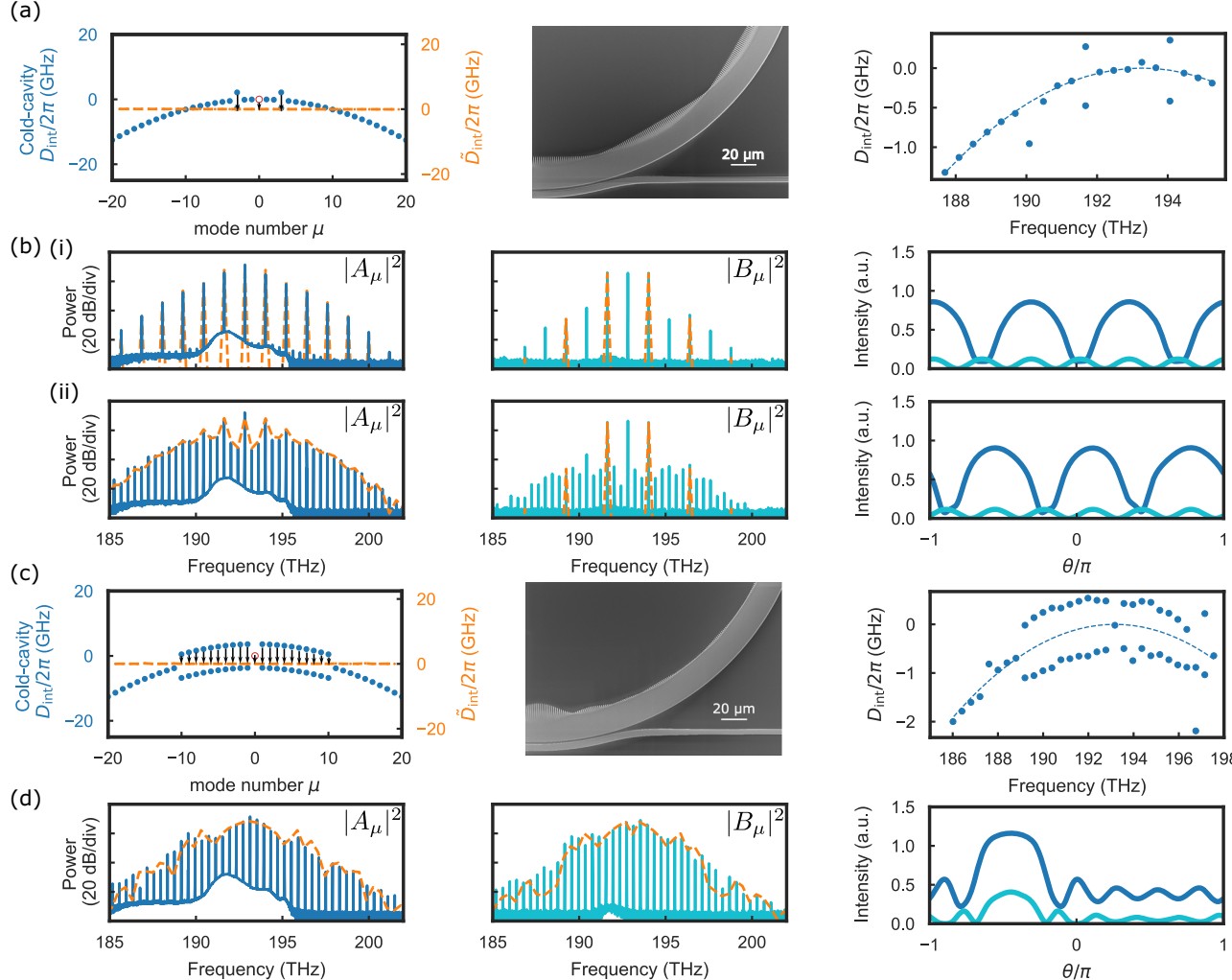

**Fig. 4 | Other designs of PhCRs that generate various types of soliton micro-combs. a** PhCRs with $\mu_s = \pm3$. The OPOs and soliton crystals generated with this design are shown in (**c**). **b** PhCRs with $\mu_s = \pm1, 2, ..., 10$. The bright soliton generated with this design is shown in (**d**). From left to right: Phase matching diagram; pulse shape; SEM images; measured $D_{int}$. **c** Spectra and pulses for design in (**a**) with (i) low and (ii) high detuning. **d** Spectra and pulses for design in (**b**). For (**c**) and (**d**), forward ($|A_\mu|^2$) and backward ($|B_\mu|^2$) spectra are plotted in solid lines, and orange dashed lines are simulation results. The forward and backward pulses are plotted in the same color as the spectra in the same direction.

emission. In exchange for controlled loss directed backward, the bandgaps on the side modes facilitate effective phase matching in the forward direction, optimizing efficiency. The bandgap-detuned excitation regime exhibits significant promise in realizing the application potential of OPOs and microcombs, spanning areas such as spectroscopy, telecommunications, computing, and quantum sensing.

## Methods
### Bandgap-detuned nanostructure definition
For the PhCR with only one split mode, we denote the average ring radius as RR, the average width as RW, and the inner corrugated wall is parametrized as $\rho_{in}(\theta) = \text{RR} - \text{RW}/2 + \rho^{\text{PHC}} \sin(2m\theta)$, where $(\rho, \theta)$ are the polar coordinates, $\rho^{\text{PHC}}$ is the amplitude of the photonic crystal and $m$ is the longitudinal order (number of wavelength) of the targeted split mode. In this paper, RR = 54.4 $\mu$m for all devices, so that the FSR is approximately 400 GHz. RW varies from 2 to 2.1 $\mu$m, so that the dispersion coefficient $D_2/2\pi \simeq -10$ MHz, $\rho^{\text{PHC}}$ is around 5 nm, and $m$ varies to split the modes at different wavelengths. For the PhCRs that generate OPOs, we can write $m$ as $m = m_0 + \mu_s$, where $m_0$ is the period that corresponds to the pump mode.

For the ring resonator with multiple photonic crystals, we design it by superposing several grating patterns in the inner wall of the

ring that are written as a parametric curve[18,37]: $\rho_{in}(\theta) = \text{RR} - \text{RW}/2 + \sum_{\mu \in \mu_s} \rho_\mu^{\text{PHC}} \sin(2(m_0 + \mu)\theta)$. We visualize the fabricated pattern by imaging a device with an exaggerated amplitude $\rho$, using a high-resolution SEM. For Fig. 2d, $\mu_s = \{-1, 1\}$ (simply denoted as $\mu_s = \pm1$ in the main text); for Fig. 4a, $\mu_s = \{-3, 3\}$; for Fig. 4a, $\mu_s = \{\pm1, \pm2, ..., \pm10\}$.

By varying the longitudinal order of the split mode and the corresponding amplitude, we can obtain arbitrary control of backscattering by adjusting the bandgaps on each mode.

### Estimation of conversion in bandgap-detuned OPOs and microcombs
The efficiency of our devices is calculated by the ratio between the comb power and the off-resonance pump power. Our simulations match the experiments well, both indicating a high forward efficiency for OPOs and combs. The efficiency in Fig. 2d is consistently above 20%, and the best forward efficiency alone can reach 48%, while in the conventional regime, the best total efficiency can reach 41% in a complex design that includes a waveguide reflector to recycle the pump field[27]. The forward efficiency for the combs with $\mu_s = \{1, -1\}$ in Fig. 3e is 37% (31% in simulation), and the backward efficiency is 15% (22% in simulation). In Fig. 4d, the forward efficiency is 24% (19% in simulation) and the backward efficiency is 14% (24% in simulation),

resulting from the more equal ratio of forward and backward emission of this design.

## Measurement of $D_{int}$

We use the standard way of measuring the integrated dispersion $D_{int}$. We send the pump into the resonator from one end of the waveguide, scan the frequency, and measure the transmission from the other end of the waveguide. Examples of transmission can be seen from the 'Mode structure' in the Supplementary Information. We fit the frequency of each mode (for the split mode, we choose the central frequency as the mode frequency) and get the FSR and $D_2$. We should note that while there are forward and backward spectra, the dispersion itself doesn't have different forward or backward components. The split mode in the dispersion has shown that there is coupling between the forward and backward fields at that mode.

## A deeper look at the coupled-mode LLEs in the stable states

Here we provide a more detailed description of the LLEs (Eqs (1) and (2)). In the LLEs, $A(\theta, \tau)$ ($B(\theta, \tau)$) describes the intracavity field in the time domain, and $A_\mu(\tau)$ ($B_\mu(\tau)$) describes the modal field in the frequency domain. They are related by $A(\theta) = \sum_\mu A_\mu e^{i\mu\theta}$ (same for $B$). The modal Kerr shift $\Delta_\mu$ can be written in the frequency domain as

$$\Delta_\mu \equiv \frac{\kappa}{2} \operatorname{Re} \left( \mathcal{F}(|A|^2 A)_\mu / A_\mu \right) = \frac{\kappa}{2} \sum_{\mu_1 + \mu_2 - \mu_3 = \mu} A_{\mu_1} A_{\mu_2} A_{\mu_3}^* / A_\mu, \quad (4)$$

where the summation symbol means the sum over all the $\mu_1, \mu_2$, and $\mu_3$ that satisfy $\mu_1 + \mu_2 - \mu_3 = \mu$.

In the stable states, $|A_\mu(\tau)|$ or $|B_\mu(\tau)|$ don't change with $\tau$, so they can be written as $A_\mu(\tau) = |A_\mu(\tau)| e^{i\eta_\mu \tau}$ and $B_\mu(\tau) = |B_\mu(\tau)| e^{i\eta_\mu \tau}$, where $\eta_\mu$ is the normalized frequency at mode $\mu$. Dividing Equation (1) by $A_\mu$, we have

$$\begin{aligned}
i\eta_\mu = & -(1+i\alpha) - iD_2\mu^2/\kappa + i\frac{\Gamma_\mu |B_\mu|}{\kappa |A_\mu|} \\
& + i \sum_{\mu_1 + \mu_2 - \mu_3 = \mu} |A_{\mu_1}||A_{\mu_2}||A_{\mu_3}^*| e^{i(\eta_{\mu_1} + \eta_{\mu_2} - \eta_{\mu_3} - \eta_\mu)\tau} / |A_\mu| \\
& + i2P_b + F\delta_{\mu,0} e^{-i\eta_\mu \tau} / |A_\mu|.
\end{aligned} \quad (5)$$

In the stable states, $\eta_\mu$ should be real and not change with $\tau$, which requires $\eta_{\mu_1} + \eta_{\mu_2} - \eta_{\mu_3} - \eta_\mu = 0$ for all $\mu_1 + \mu_2 - \mu_3 = \mu$ and $\eta_0 = 0$. It's easy to see that $\eta_\mu$ must be linear with $\mu$ to make the states stable. Note that $\eta_\mu \kappa/2 = \tilde{D}_{int}(\mu)$, so we have proved $\tilde{D}_{int}(\mu) = \mu\delta\omega_{rep}$ in the main text, and we can write $\eta_\mu = 2\mu\delta\omega_{rep}/\kappa$. The linearity of $\tilde{D}_{int}(\mu)$ or $\eta_\mu$ with $\mu$ indicates the same line spacing between the comb (or OPO) lines in the stable states.

In the time domain $A(\theta, \tau) = \sum_\mu |A_\mu| e^{i\mu(\theta + 2\delta\omega_{rep}\tau/\kappa)}$. Thus, in simulation, if $\delta\omega_{rep}$ is not zero (usually caused by $\Gamma_\mu \neq \Gamma_{-\mu}$), we can see that $A(\theta, \tau)$ is moving at a constant speed as $\tau$ evolves.

## Data availability

Source data have been uploaded. Additional information is available from the corresponding author upon request. Source data are provided with this paper.

## Code availability

All the simulation codes used in this study are available from the corresponding author upon request.

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

## Acknowledgements

We thank Grant Brodnik and Lindell Williams for reviewing the manuscript. This work is funded by the DARPA PIPES program HR0011-19-2-0016, the DARPA QuICC program FA8750-23-C-0539, the DARPA NaPSAC program, AFOSR FA9550-20-1-0004 Project Number 19RT1019, NSF Quantum Leap Challenge Institute Award OMA-2016244, and NIST. This work is a contribution of NIST and is not subject to US copyright. Mention of specific companies or trade names is for scientific communication only and does not constitute an endorsement by NIST.

## Author contributions

Y.J. performed the conception, design, and theoretical analysis. Y.J., J.Z., T.B., and I.D. collected the data. Y.J. analyzed the data. D.C. fabricated the devices. Y.J., E.L., and S.B.P. wrote the paper. E.L. and S.B.P. contributed to the theoretical understanding. S.B.P. supervised the findings of this work. All authors reviewed and helped shape the research, analysis, and manuscript.

## Competing interests

The authors declare no competing interests.
