## [Transparent Peer Review file · Nature Communications]

The bandgap-detuned excitation regime in photonic-crystal resonators

Corresponding Author: Mr Yan Jin

Version 0:

Reviewer comments:

Reviewer #1

(Remarks to the Author)

This manuscript reports some theoretical and experimental studies of nonlinear dynamics (OPO and soliton micro-combs) in photonic-crystal resonators in the so-called “bandgap-detuned” excitation regime. In this regime the grating induced coupling between forward- and backward-propagating fields is tuned to be at a higher order mode, relative to the pump. This is in contrast to the “conventional” scheme, where this coupling occurs at the pump mode. According to the Authors, the main advantages of this scheme are the higher efficiency of frequency conversion (compared to the conventional scheme) and the added versatility in “arbitrary control of backscattering”. Both claims I found to be poorly justified by the presented data. Specifically:

1. I am struggling to find any substantial discussion in text related to the “high efficiency” statement. There are some statements about “notably higher threshold power” observed in the “conventional” scheme, and comparison of the sideband powers between the two regimes in simulation (page 4). But I could not find any corresponding data or plots. Given that this is one of the two important advantages of the new scheme, as explained in the introductory part, I was expecting a bit more detailed discussion around that.

2. I do not understand what the Authors mean by “arbitrary control of backscattering”, as this claim is not backed by any data. I can see from Fig. 1a and 1b that the structure of a steady state is different for the two schemes, but one can hardly call this an “arbitrary control”, unless I am missing something? (See also more on this point later in my comments 5 and 6).

Overall, I find this manuscript very difficult to follow by someone who is not an expert in the field of micro-resonators. Many claims are made without proper explanations, and the text is full of technical jargon. In particular:

3. The main idea of this work is to create a coupling between forward- and backward-propagating modes of a resonator in one particular mode only (labelled μ_s). It is never explained how this is achieved in practice. I can see how by choosing a proper period of the inner grating, one can resonantly enhance this coupling for a given mode (although, this is only my guess, I could not find a clarification in the manuscript). But how realistic is the assumption that the coupling for all other modes is negligible (throughout this work it was assumed to be in the form of the Kronecker’s delta-function)? But surely, there will be some backscattering from the grating in all other modes? And I can see e.g. in Fig. 1f that there is a significant back-scattered component of the pump (which is obviously not predicted by the theory). Hence I wonder how accurate is this assumption?

4. For the OPO setup discussed in Fig. 1, how the parametric interaction is arranged such that only the specific modes with $\mu=+\mu_s$ and $\mu=-\mu_s$ are excited? Is this done by arranging the phase-matching condition? I would like to see a more detailed discussion.

5. I do not understand what is plotted in Figs. 1a and 1b. According to the text, these are “benchmark simulations of OPO states”. But what do the Authors mean by “benchmark simulations”? Are these stationary solutions of the system (1), (2) for some specific value of the pump amplitude? Would the structure of these solutions be qualitatively similar for all pump amplitudes F , or would it change? Again, a clarification is required. Also, to add to the confusion, the excited sidebands in Figs. (1) and (2) are at $\mu=+14$ and $\mu=-14$. However, the plot in Fig. 1c shows the arranged coupling between forward- and backward-propagating modes for $\mu_s=10$. Is this a different geometry?

6. The discussion about the “integrated dispersion” and mode-locked states, and the related control of field amplitudes, on page 4 is very confusing. First, there is a statement that the “relative field amplitudes A and B are influenced by the interplay between the bidirectional coupling strength ϵ and the Kerr nonlinearity”. If I understand the Authors correctly, this is due to the parabolic dispersion $\sim D^2$ being effectively compensated by the coupling and Kerr terms. I can see, in principle, how this balance will require different field profiles, depending on the overall power and the specific bidirectional coupling arrangement. But I cannot see how one can utilize this for controlling “each of the field amplitudes A or B with each bandgap Γ ”, as stated later. I would like to see a more detailed discussion here, backed up by further data. It is hard to make any conclusions just from one solution presented!

7. I do not understand what is plotted in Fig. 2a and how this is related (or not related) to the efficiency of the OPO process? What significance $\Delta\Omega$ has? Also, the Author state that this map was “not computed via LLE simulations, but by exploring the steady state...” Presenting more details on the corresponding numerical/theoretical investigation would be useful!

8. On page 4, there is also a discussion about the structure of the gain in two different schemes, with a claim that the bandgap-detuned regime is characterized by the “dominance of the forward gain”, as opposed to the “conventional” scheme. I do not understand how the Authors came to this conclusion. Is this a mere observation, or was there some detailed gain analysis performed? I would like to see more details!

9. An excessive number of introduced parameters makes this manuscript difficult to follow. For example, there are three parameters β , Γ , and ϵ , which are all related to one and the same physical characteristic: the strength of bidirectional coupling. And all three parameters are used interchangeably throughout the manuscript.

To summarize, in my opinion, this manuscript would be better suited for a more specialized journal, and even then it would require some substantial revisions.

Reviewer #3

(Remarks to the Author)

In this manuscript, Y. Jin et al. thoroughly explore nonlinear interactions in photonic-crystal resonators, with a particular focus on the bandgap-detuned regime. The authors experimentally and theoretically demonstrate the generation of optical parametric oscillations and frequency combs. Their approach distinguishes itself from conventional methods by introducing an additional degree of freedom for dispersion engineering, which is essential for wideband comb generation. I believe this work has the potential to make a significant impact on the photonics and nonlinear optics communities by opening new pathways for nonlinear engineering.

The paper is technically well-written, with clear schematics and a presentation of results. It is well-suited for the target journal and relevant to its readership, and I recommend it for publication. However, I suggest addressing a few minor points to ensure clarity for all readers:

- 1) In Fig. 1(f), it looks like the pump spectrum (at $\mu = 0$) for the forward direction appears smaller than the backward one. I understand reflection at the edge of the device may introduce the unwanted backscattering at $\mu_s = -10$. However, it is quite puzzling that the forward pump seems slightly lower than the backward one. Could this result from a calibration issue in estimating power? Please clarify this point.
- 2) Perhaps I missed it from the text, but I could not locate specific values for the pump power used in Fig. 1(c). Could the authors specify the values of F they used for the ‘low’ and ‘high’ pump power cases?
- 3) While the paper is well-written, some sections need to be reorganized. For instance, μ_s is introduced on page 4 without definition, and only later is it defined. Additionally, it is unclear how efficiency is calculated on page 6 and in Fig 2(d). Similarly, I do not understand what $\Gamma_{\mu} = \Gamma \Delta \mu$ means in the paragraph explaining the bandgap-detuned regime on page 4. Specifically, if I'm not mistaken, Δ seems to be not defined throughout the text. The authors might consider adding a table of nomenclature in the Methods section, as the text includes many variables, which could make it challenging for readers to follow.

Version 1:

Reviewer comments:

Reviewer #1

(Remarks to the Author)

Reviewer #3

(Remarks to the Author)

The authors successfully addressed my comments, and I think this paper is much clearer now. I recommend this work for publication in Nature Communications.

That said, I concur with Referee 1 that the clarity of the manuscript could be improved. The current presentation, particularly the writing style and figure descriptions, may be overly technical for a broader readership. This might contribute to the impression that the paper is better suited for a more specialized journal. One concrete suggestion is to improve the figure captions—especially for Figs. 1(a) and 1(b). I assume the upper panels in these figures show the ‘backward’ spectra in the presence of a ‘forward’ pump, but this is not clearly stated. Furthermore, it is unclear why the comb lines at $\mu = \pm 10$ are not visible in the ‘forward’ direction in Fig. 1(a), yet appear in Fig. 1(b). I assume this is due to phase mismatch introduced by mode splitting, but this should not be left to the reader’s inference.

Additionally, the lower panels of Figs. 1(a) and 1(b) are not explained at all. Fig. 1(c) appears to be central to explaining the underlying physics of the selective comb generation seen in Fig. 1(b), yet the caption does not sufficiently guide the reader. It is also unclear what dispersion is plotted in Fig. 1(c). While I presume it shows the forward mode dispersion, this is confusing, especially given that the preceding panels display the backward spectra. These figure transitions—and the corresponding transitions in the text—would benefit from improved explanation and clearer structure to help readers follow the logical flow of the work.

To summarize, the paper demonstrates strong technical rigor and presents a novel contribution. However, this same rigor can at times hinder accessibility, making it difficult for readers—especially those outside the immediate subfield—to fully grasp the main ideas. It is a double-edged sword. I suggest the authors make more effective use of the Supplementary Information to support detailed content, while ensuring that the main messages are clearly conveyed in the main text. Striking a better balance between technical depth and clarity of presentation would allow the significance of the work to be more broadly appreciated.

The images or other third party material in this Peer Review File are included in the article’s Creative Commons license, unless indicated otherwise in a credit line to the material. If material is not included in the article’s Creative Commons license and your intended use is not permitted by statutory regulation or exceeds the permitted use, you will need to obtain permission directly from the copyright holder.

Author response letter

Nature Communications manuscript NCOMMS-24-42483-T
“The bandgap-detuned excitation regime in photonic-crystal resonators”

We thank all the referees for their thoughtful comments. Here, we provide responses to the referees in **blue text**, and **bold blue text** specifically denotes changes to the manuscript that we have made to address the referee comments. The changes in the main text are marked in **red**.

Reviewer #1 (Remarks to the Author):

This manuscript reports some theoretical and experimental studies of nonlinear dynamics (OPO and soliton micro-combs) in photonic-crystal resonators in the so-called “bandgap-detuned” excitation regime. In this regime the grating induced coupling between forward- and backward-propagating fields is tuned to be at a higher order mode, relative to the pump. This is in contrast to the “conventional” scheme, where this coupling occurs at the pump mode. According to the Authors, the main advantages of this scheme are the higher efficiency of frequency conversion (compared to the conventional scheme) and the added versatility in “arbitrary control of backscattering”. Both claims I found to be poorly justified by the presented data. Specifically:

1. I am struggling to find any substantial discussion in text related to the “high efficiency” statement. There are some statements about “notably higher threshold power” observed in the “conventional” scheme, and comparison of the sideband powers between the two regimes in simulation (page 4). But I could not find any corresponding data or plots. Given that this is one of the two important advantages of the new scheme, as explained in the introductory part, I was expecting a bit more detailed discussion around that.

Thanks to the referee for pointing this out. Indeed, the high conversion efficiency of the bandgap-detuned regime is an important breakthrough, and we have measured and evaluated the conversion efficiency in detail. However, we can now see that this information did not come through clearly in the paper. **We have made several revisions in the paper to address this point:**

- **For several OPO and comb states reported in the paper, we have added sentences that indicate the measured and predicted efficiency. Moreover, in the Methods section we have a description of the measured and predicted conversion efficiency.** In “Estimation of conversion in bandgap-detuned OPOs and microcombs.” of Methods, “The efficiency in Figure 2(d) is consistently above 20%, and the best forward efficiency alone can reach 48%, while in the conventional regime, the best total efficiency can reach 41% in a complex design that includes a waveguide reflector to recycle the pump field [27]. The forward efficiency for the combs with $\mu_s = \{1, -1\}$ in Figure 3(e) is 37%, and the backward efficiency is 15%. In Figure 4(d), the forward efficiency is 24 % and the backward efficiency is 14 %.”

- **We have added information in the initial theory section to quantify OPO and microcomb conversion efficiency predicted from the modeling.**
- **In approximately line 132, we have added a word “Methods”.**
- **In approximately line 194, we have added a sentence “See Methods for more details about the OPO conversion efficiency.”**

2. I do not understand what the Authors mean by “arbitrary control of backscattering”, as this claim is not backed by any data. I can see from Fig. 1a and 1b that the structure of a steady state is different for the two schemes, but one can hardly call this an “arbitrary control”, unless I am missing something? (See also more on this point later in my comments 5 and 6).

We thank the referee for pointing out this confusing part of our paper. **We have corrected this part in the abstract, noting the arbitrary, mode by mode control of the backscattering rate. We have also indicated how mode-by-mode control is important in the initial theoretical section.**

We have further clarified the paper by changing the discussion about arbitrary control. **In the 3rd paragraph of “Bandgap-detuned nanostructure definition” in the Methods section, we have added a sentence “By varying the longitudinal order of the split mode and the corresponding amplitude, we can obtain arbitrary control of backscattering by adjusting the bandgaps on each mode.”**

We have also answered the questions from the referee’s comments 5-6 in detail; see more below. From “Mode structures” in Methods around line 351, we have presented the ability to adjust the bandgap on many different modes.

Overall, I find this manuscript very difficult to follow by someone who is not an expert in the field of micro-resonators. Many claims are made without proper explanations, and the text is full of technical jargon. In particular:

We can somewhat understand this point, and we appreciate the opportunity to clarify our paper. While we think the underlying treatment of the LLE formalism for the resonator nonlinear dynamics is standard within the broad context of soliton microcomb literature, we do understand that the level of sophistication of the field is rather high and less accessible. **Therefore, we have simplified the LLE formalism in the paper, removed as many parameters as possible, provided added description of the parameters, and moved details into the methods.**

3. The main idea of this work is to create a coupling between forward- and backward-propagating modes of a resonator in one particular mode only (labelled μ_s). It is never explained how this is achieved in practice. I can see how by choosing a proper period of the inner grating, one can resonantly enhance this coupling for a given mode (although, this is

only my guess, I could not find a clarification in the manuscript). But how realistic is the assumption that the coupling for all other modes is negligible (throughout this work it was assumed to be in the form of the Kronecker's delta-function)? But surely, there will be some backscattering from the grating in all other modes? And I can see e.g. in Fig. 1f that there is a significant back-scattered component of the pump (which is obviously not predicted by the theory). Hence I wonder how accurate is this assumption?

We thank the referee for pointing out these areas that we can clarify in the paper. There have now been dozens of papers on OPOs and microcombs in PhCRs from nearly 10 different groups, so there is quite a depth of information in the literature. **We have made several changes in the paper to clarify.** In PhCRs, the resonator waveguide contains a periodic modulation of the waveguide width with a variable modulation amplitude. The period of the modulation corresponds directly to the resonator mode number and the amplitude controls the bandgap of that mode, and we address separate modes simply by changing the period; **see modified description in the main text.**

With regard to realizing the forward/backward coupling in practice, **we have added several sentences in “Bandgap-detuned nanostructure definition” of the Methods section** to explain photonic-crystal resonators with both a single period to address a single mode and several periods to address several modes.

Of course, we have carefully calibrated that PhCRs with multiple periods can independently address separate modes. Both in previous papers (Lucas Nature Photonics “Tailoring microcombs with inverse-designed, meta-dispersion microresonators”) and in this paper, we have calibrated the mode spectrum. **We have now added information in the main text and the methods** that directly states any inadvertent bandgap that forms on incorrect modes. **In “Mode structures” of the Methods section, we have presented the mode structures for all the different split mode structures mentioned in this paper. It is clear that the split modes don't affect other modes and the coupling is negligible for all other modes.** This clarifies the assumption about the Kronecker delta function.

With regard to the data in Fig. 1f, we thank the referee for raising this as it deserves clarification. The reflected component of the pump is not from the splitting, but from the reflection of the angled facets at both ends of the on-chip waveguide. Since light reflection from the facet does not suffer loss of coupling to the waveguide, it appears larger. Thus, the reflected pump comes from 2 sources: the front (input) and rear (output) sides of the waveguide, while the reflected comb line between 180 THz and 190 THz in Fig. 1f only comes from the end side of the waveguide. **We added a note about this in the figure caption. We have also added sentences approximately in around line 154 to clarify this, “The backward comb line at $\mu = 0$ is particularly strong because the pump is reflected at both the front (input) and rear (output) facets of the chip, while the comb line at $-\mu_s$ is only reflected at the rear facet of the chip. ...”**

For other questions raised by the referee specific to this point:

The main idea of this work is to create a coupling between forward- and backward-propagating modes of a resonator in one particular mode only (labelled μ_s)

Actually for the OPO part, we only create one grating coupling in one particular mode. You will only see one sinusoidal grating, as shown in the SEM figure of Fig. 1(e); but for the combs, we create multiple μ_s by superposing several grating patterns (See the SEM figures in Fig. 3(d) and Fig.4 (a)(c), and the way we design it in the second paragraph of “Bandgap-detuned nanostructure definition” in the Methods part). In short, we vary the integer variable $m=m_0+\mu_s$ in the mask design so that we can split different mode μ_s . **The corresponding mode structures can be found in “Mode structures” of the Methods section.**

But surely, there will be some backscattering from the grating in all other modes?

No. If we split the mode at μ_s , it will not affect other modes. The reflection seen is from the rear side of the waveguide. **See “Mode structures” of the Methods section.**

Hence I wonder how accurate is this assumption?

Our assumption, really the treatment of this important aspect in the work, is accurate. The new data we have provided clarifies and further confirms our treatment of the PhCR mode splitting structure.

4. For the OPO setup discussed in Fig. 1, how the parametric interaction is arranged such that only the specific modes with $\mu=+\mu_s$ and $\mu=-\mu_s$ are excited? Is this done by arranging the phase-matching condition? I would like to see a more detailed discussion.

In experiment, it's arranged by adding a grating $\rho_{phc} \sin(2m\theta)$ to the inner wall of the ring (see Methods), where $m = m_0 + \mu_s$ and ρ_{phc} is the grating amplitude. By varying m in the design of the ring, we can change the μ_s ; and by varying ρ_{phc} , we are able to change the amplitude of Γ_μ in Eqs (1) and (2).

In simulation, we calculated the range of Γ_μ and D_2 where the OPOs can be generated (or the phase matching condition is achieved), and then design the devices accordingly. Fig. 2 describes how the experiment and simulation match successfully.

We have clarified this point in the subsection “Bandgap-detuned nanostructure definition” of the Methods section.

5. I do not understand what is plotted in Figs. 1a and 1b. According to the text, these are “benchmark simulations of OPO states”. But what do the Authors mean by “benchmark simulations”? Are these stationary solutions of the system (1), (2) for some specific value

of the pump amplitude? Would the structure of these solutions be qualitatively similar for all pump amplitudes F , or would it change? Again, a clarification is required. Also, to add to the confusion, the excited sidebands in Figs. (1) and (2) are at $\mu=+14$ and $\mu=-14$. However, the plot in Fig. 1c shows the arranged coupling between forward- and backward-propagating modes for $\mu_s=10$. Is this a different geometry?

We will answer these questions one by one. In the following, we repeat the referee question and show our responses.

what do the Authors mean by “benchmark simulations”? Are these stationary solutions of the system (1), (2) for some specific value of the pump amplitude?

By benchmark, we mean these indicate the fundamental improvement offered by the bandgap-detuned regime. **Around line 117 we have added a sentence “to indicate the fundamental improvement” to make it clear.**

Yes, these are stationary solutions. **We have now added the specific pump amplitude used to the text. Around line 115, “We set F as 2.6 and 2 in the simulation for Figure 1(a) and Figure 1(b), respectively.”**

Would the structure of these solutions be qualitatively similar for all pump amplitudes F , or would it change?

Yes, the **stationary** results in Fig. 1a and 1b are similar for all pump amplitude. With a large F , the spectra will not be stable anymore. **We have clarified this in the text. We have marked $F=2$ in OPOs in the main text, around line 307 of the newly added subsection “Calculation of parametric gain” in the Methods section, we have the sentence “We should note that the existence map only provides the region where the gain is greater than 0, and it doesn’t predict the stability or efficiency of the OPOs which need to be simulated directly by Eqs. (1-2). The OPO pattern generated with reasonable F and Γ/k is similar; however, with a large F (usually greater than 2.5), the OPOs tend to be unstable.”**

Also, to add to the confusion, the excited sidebands in Figs. (1) and (2) are at $\mu=+14$ and $\mu=-14$. However, the plot in Fig. 1c shows the arranged coupling between forward- and backward-propagating modes for $\mu_s=10$. Is this a different geometry?

We thank the referee for pointing out this confusion. **We changed spectra and pulses to the case with $\mu_s = 10$ in Figure 1(b) so that the comparison is clearer and the panels in this figure are more consistent.**

6. The discussion about the “integrated dispersion” and mode-locked states, and the

related control of field amplitudes, on page 4 is very confusing. First, there is a statement that the “relative field amplitudes A and B are influenced by the interplay between the bidirectional coupling strength ϵ and the Kerr nonlinearity”. If I understand the Authors correctly, this is due to the parabolic dispersion $\sim D_2$ being effectively compensated by the coupling and Kerr terms. I can see, in principle, how this balance will require different field profiles, depending on the overall power and the specific bidirectional coupling arrangement. But I cannot see how one can utilize this for controlling “each of the field amplitudes A or B with each bandgap Γ ”, as stated later. I would like to see a more detailed discussion here, backed up by further data. It is hard to make any conclusions just from one solution presented!

Here the referee is asking for a more detailed discussion about a specific point. We thank the referee for pointing this out. Our intent is to express that this conclusion is made from the theoretical discussion from the previous paragraph, which builds up our framework instead of “just from one solution presented.” **We rephrase it with the last sentence from the previous paragraph: since $D_{int}(\mu)$ and $D_2 \mu^2$ are fixed, we vary the bandgap Γ_{μ} (or ϵ_{μ}) to change the value of Kerr shift Δ_{μ} , which is a function of A_{μ} or B_{μ} , and thus controlling A_{μ} or B_{μ} .**

This conclusion is also verified by the simulation or experiments all over the paper:

- (1) In Fig. 1(f), only the mode at $\mu_s=10$ is split (only bandgap $\Gamma_{10}>0$, and other bandgaps are 0), and since there is backward reflection, A_{10} is smaller than $A_{(-10)}$; however, it’s greater than the other modes without bandgaps (-10 excluded). The simulation result in Fig. 1(f) also verifies that.
- (2) In Fig. 3 where the modes at ± 1 are split, amplitudes A and B at $\mu_s=\pm 1$ are much stronger than the other modes. However, because there is a slight difference between Γ_{1} and Γ_{-1} ($\Gamma_{1}>\Gamma_{-1}$), the amplitudes A, B at $+1$ and -1 are also different ($A_{1} > A_{-1}$).
- (3) Results in Fig. 2 and 4 all show that the amplitudes at the split modes are stronger than the other un-split modes (except the modes at the opposite position of the split modes).

7. I do not understand what is plotted in Fig. 2a and how this is related (or not related) to the efficiency of the OPO process? What significance $\Delta\Omega$ has? Also, the Author state that this map was “not computed via LLE simulations, but by exploring the steady state...” Presenting more details on the corresponding numerical/theoretical investigation would be useful!

Fig. 2a is an existence map, i.e., the region of β and μ_s where the OPOs can be generated, ie. phase matching occurs. It is not directly determinative of the efficiency of the OPO process. **We have added information in the main text to clarify the meaning of this figure. Around line 168, we have added a sentence “A detailed derivation can be found in the Methods section”. Around line 321 of the Methods section, we have**

added a sentence “We should note that the existence map only provides the region where the gain is greater than 0, and it doesn’t predict the stability or efficiency of the OPOs which need to be simulated directly by Equations. (1-2)”.

DeltaOmega is the difference between the repetition rate of the generated combs and the FSR of the resonances in the cold cavity. **We have modified a sentence in the main text to make this clearer. Around line 100 we have added a sentence “where $\delta\omega_{\text{rep}}/2\pi = \text{frep} - \text{FSR}$ is the difference between the repetition rate of the comb (or line spacing) and the free spectral range which is measured in the cold cavity.” (meaning)**

In our theory, we predicted that the comb line frequency is shifted from the resonance by $\mu \cdot \Delta\omega$ (i.e., \tilde{D}_{int}) at mode μ , and our experiment matches the prediction well. This indicates that we have precise control of nonlinear interaction in the bandgap detuned regime. **We have added a sentence in the paper to emphasize that. Around line 182, we have added a sentence “It validates our prediction of the behaviour of effective dispersion \tilde{D}_{int} in Figure 3 and showcases the precise control of nonlinear interaction in the bandgap-detuned regime”. (significance)**

We have presented the detailed numerical investigation in the subsection “Calculation of parametric gain” of the Methods section.

8. On page 4, there is also a discussion about the structure of the gain in two different schemes, with a claim that the bandgap-detuned regime is characterized by the “dominance of the forward gain”, as opposed to the “conventional” scheme. I do not understand how the Authors came to this conclusion. Is this a mere observation, or was there some detailed gain analysis performed? I would like to see more details!

We did perform a gain analysis to create this information in the original draft. **To clarify this issue, we have added information in “Calculation of parametric gain” of the Methods section. And we have added a sentence (around line 125) in the main text to clarify this information and point the reader to the Methods for more details. Moreover, we have clarified in the paper that Ref [33] has more details.**

9. An excessive number of introduced parameters makes this manuscript difficult to follow. For example, there are three parameters beta, Gamma, and epsilon, which are all related to one and the same physical characteristic: the strength of bidirectional coupling. And all three parameters are used interchangeably throughout the manuscript.

We understand this comment. **To address it, we have reduced the number of variables names used in the paper, particularly the ones the referee mentioned. We removed beta in favor of keeping Gamma and epsilon.** Although those two look similar, they have different physical meanings. Gamma is simply the bandgap of the split mode, while epsilon not only shows the frequency shift from the original resonance, but also which direction that the frequency shifts. **Therefore, we have also added a sentence in the paper that explains the difference between those parameters. Around line 96, “The**

different signs of ϵ indicate the frequency of the comb line at μ is greater ($\epsilon_{\mu} > 0$) or smaller ($\epsilon_{\mu} < 0$) than the center of two resonances at the split mode”

To summarize, in my opinion, this manuscript would be better suited for a more specialized journal, and even then it would require some substantial revisions.

We thank the referee for pointing out the confusion and problems in this paper. We have addressed all of the points that the referee raised, and we feel that with these clarifications the paper is appropriate for Nature Communications.

Reviewer #2 (Remarks to the Author):

In this manuscript, Y. Jin et al. thoroughly explore nonlinear interactions in photonic-crystal resonators, with a particular focus on the bandgap-detuned regime. The authors experimentally and theoretically demonstrate the generation of optical parametric oscillations and frequency combs. Their approach distinguishes itself from conventional methods by introducing an additional degree of freedom for dispersion engineering, which is essential for wideband comb generation. I believe this work has the potential to make a significant impact on the photonics and nonlinear optics communities by opening new pathways for nonlinear engineering.

We thank the referee for carefully considering our paper and the thoughtful comments for improvement.

The paper is technically well-written, with clear schematics and a presentation of results. It is well-suited for the target journal and relevant to its readership, and I recommend it for publication. However, I suggest addressing a few minor points to ensure clarity for all readers:

The responses below show how we have addressed the referee’s comments.

1) In Fig. 1(f), it looks like the pump spectrum (at $\mu = 0$) for the forward direction appears smaller than the backward one. I understand reflection at the edge of the device may introduce the unwanted backscattering at $\mu_s = -10$. However, it is quite puzzling that the forward pump seems slightly lower than the backward one. Could this result from a calibration issue in estimating power? Please clarify this point.

Referee 1 also commented on this point, so we have taken particular care to address it with modifications in the paper.

Indeed, as the referee suggests this detail is confusing and it is explained by the experimental configuration. The backscattered component of the pump is not any photonic crystal but from the reflection of the chip facets. The front (input) facet reflection is particularly strong because the pump has not been attenuated

In the main text description of this figure, we have added a clear discussion of these effects so that the reader can better understand the data.

We use the lensed fiber to couple the pump into waveguide, and there is already reflection at the start of the waveguide. When the pump is coupled into the waveguide, it will propagate and generate combs in the ring cavity. The generated forward combs and the residue pump then propagate forward and are reflected by the end side of the waveguide. Thus, the back-scattered pump comes from 2 sources: the front (input) and rear (output) sides of the waveguide, while the back-scattered comb line between 180 THz and 190 THz in Fig. 1f only comes from the end side of the waveguide.

The reflected power at the start side of the waveguide is the strongest because there is only pump power at this point, so you can see a much stronger reflection in the pump than the comb line in the backward direction.

We also tested the reflected power of the pump when not generating combs (our pump is off resonance of the ring cavity), and the reflected pump is significant. We should note that the reflected pump at this point has nothing to do with the efficiency calculation.

In our simulation, we only consider the fields inside the ring cavity but not the reflection of the waveguide ends. That's why we don't have the back-scattered pump or comb line in the simulation.

Around line 152, we have added sentences to clarify these points.

2) Perhaps I missed it from the text, but I could not locate specific values for the pump power used in Fig. 1(c). Could the authors specify the values of F they used for the 'low' and 'high' pump power cases?

In the revised manuscript, we have specified the values of F for the 'low' and 'high' pump power:

In the case of 'low' pump power, F is almost 0 and it just plots the dispersion in blue. We have provided this information around line 129: **"For $F = 0$, we plot the cold cavity effective dispersion (blue) ..."**

Around line 115, we have clarified that F is 2 for the high-power case: **"We set F as 2.6 and 2 in the simulation for Figure 1(a) and Figure 1(b), respectively."**

3) While the paper is well-written, some sections need to be reorganized. For instance, μ_s is introduced on page 4 without definition, and only later is it defined. Additionally, it is unclear how efficiency is calculated on page 6 and in Fig 2(d). Similarly, I do not

understand what $\Gamma_\mu = \Gamma \delta_\mu$ means in the paragraph explaining the bandgap-detuned regime on page 4. Specifically, if I'm not mistaken, δ seems to be not defined throughout the text. The authors might consider adding a table of nomenclature in the Methods section, as the text includes many variables, which could make it challenging for readers to follow.

We thank the referee for pointing the typo out. Here δ is the Kronecker delta. It's actually $\Gamma_\mu = \Gamma \delta_{\{\mu, \mu_s\}}$, which means $\Gamma_\mu = 0$ if μ is not equal to the split mode number μ_s , and $\Gamma_\mu = \Gamma$ when μ is the μ_s . **Around line 108 we have clarified this by saying “where $\mu_s \neq 0$ is the split mode number and δ is the Kronecker delta”.**

We have reduced the number of variables throughout the paper. In particular, we have removed the normalized variables d_2 and β .

As suggested by the referee, in the methods we provided a clear description (“A deeper look at the coupled-mode LLEs in the stable states”) of the LL equations that we use so that the interested reader can easily track with the underlying formalism.

Author response letter

Nature Communications manuscript NCOMMS-24-42483B
“The bandgap-detuned excitation regime in photonic-crystal resonators”

We thank all the referees for their thoughtful comments. Here, we provide responses to the referee(s) in **blue text**, and **bold blue text** specifically denotes changes to the manuscript that we have made to address the referee comments. The changes in the main text are marked in **red**.

Reviewer #3 (Remarks to the Author):

The authors successfully addressed my comments, and I think this paper is much clearer now. I recommend this work for publication in Nature Communications.

We thank the referee for carefully considering our paper and the thoughtful comments for improvement. We have modified the caption, main text and supplementary materials so that it's clearer to the readers. We have put some sections in Methods to an additional supplementary information document.

That said, I concur with Referee 1 that the clarity of the manuscript could be improved. The current presentation, particularly the writing style and figure descriptions, may be overly technical for a broader readership. This might contribute to the impression that the paper is better suited for a more specialized journal. One concrete suggestion is to improve the figure captions—especially for Figs. 1(a) and 1(b).

We thank the referee for pointing this out and providing us with the concrete suggestion. We have modified the captions of Figure 1 as well as the main text.

I assume the upper panels in these figures show the ‘backward’ spectra in the presence of a ‘forward’ pump, but this is not clearly stated.

We have added “the combs propagate in the backward direction against the pump.” for Figure 1(a) and “the combs mainly propagate in the forward direction, i.e, the same direction as the pump.” for Figure 1(b) in the caption.

Furthermore, it is unclear why the comb lines at $\mu = \pm 10$ are not visible in the ‘forward’ direction in Fig. 1(a), yet appear in Fig. 1(b). I assume this is due to phase mismatch introduced by mode splitting, but this should not be left to the reader's inference.

We thank the referee for pointing this out. In the main text, we have added **“Due to a stronger backward pump field in the conventional regime, the OPOs only propagate in**

the backward direction. In the bandgap-detuned regime, because of the coupling between the forward and backward fields at $\mu_s=10$ induced by the bandgap, there is a backward-propagating OPO line only at $\mu = 10$." around line 123.

Additionally, the lower panels of Figs. 1(a) and 1(b) are not explained at all.

We thank the referee for pointing this out. **In the main text, we have added "The pulse intensities in Figure 1(a) and (b) present the corresponding $|A(\theta)|^2$ (dark blue) and $|B(\theta)|^2$ (light blue) normalised to the pumping power F^2 . They all have 10 oscillations in a round trip because the OPO lines emerge at $\mu = \pm 10$ for both regimes. Comparing these two normalised intensity profiles, it is clear that the bandgap-detuned regime has a higher efficiency"** around line 128.

Fig. 1(c) appears to be central to explaining the underlying physics of the selective comb generation seen in Fig. 1(b), yet the caption does not sufficiently guide the reader.

We thank the referee for pointing this out. We have rephrased the caption for Figure 1(c): **"Effective integrated dispersion \tilde{D}_{int} for $\mu_s = 10$ in the cold cavity (blue) and hot cavity (orange). As we increase the pump power, the modal Kerr shift $\Delta\mu$ compensates the original cold cavity dispersion D_{int} (blue) and the bandgap-induced shift ϵ_{μ_s} , and pushes the optical modes to a straight line (orange) which achieves the phase-matching condition and generates the stable OPO."**

It is also unclear what dispersion is plotted in Fig. 1(c). While I presume it shows the forward mode dispersion, this is confusing, especially given that the preceding panels display the backward spectra.

We measure the dispersion in the forward direction in the standard way. **We have added a section in Methods "Measurement of D_{int} " to show how we measure the dispersion.**

Indeed, there are forward and backward spectra for the bandgap-detuned regime. However, we should note that **the dispersion itself doesn't have different forward or backward components. We also point this out in Methods.** At the split mode ($\mu_s=10$) in Figure 1(c) there are 2 modes or resonances, which is a result of the coupling between the forward and backward fields. Therefore, both the forward and backward OPO or comb lines are both generated at that split mode.

We have added "The split mode at $\mu_s = 10$ with 2 resonances indicates that there is a coupling between the forward and backward fields, which results in generating both the forward and backward OPO or comb lines at that mode. In the experiment, we measure $D_{int}(\mu)$ in the forward direction; see Methods for more details." around line 134.

These figure transitions—and the corresponding transitions in the text—would benefit from improved explanation and clearer structure to help readers follow the logical flow of the work.

We thank the referee for pointing these problems out. We have modified the caption and the main text so that they are clearer to the readers.

To summarize, the paper demonstrates strong technical rigor and presents a novel contribution. However, this same rigor can at times hinder accessibility, making it difficult for readers—especially those outside the immediate subfield—to fully grasp the main ideas. It is a double-edged sword. I suggest the authors make more effective use of the Supplementary Information to support detailed content, while ensuring that the main messages are clearly conveyed in the main text. Striking a better balance between technical depth and clarity of presentation would allow the significance of the work to be more broadly appreciated.

We thank the referee for pointing this out. We have put some details into the Supplementary Information and added one section in Methods so that we convey the messages more clearly to the readers.